# Cottontail Rabbit Papillomavirus (CRPV) Related Animal Models for Head and Neck Cancer Research: A Comprehensive Review of the Literature

**DOI:** 10.3390/v16111722

**Published:** 2024-10-31

**Authors:** Michael Bette, Robert Mandic

**Affiliations:** 1Institute of Anatomy and Cell Biology, Philipps-Universität Marburg, 35037 Marburg, Germany; 2Department of Otorhinolaryngology, Head and Neck Surgery, University Hospital Marburg, Philipps-Universität Marburg, 35033 Marburg, Germany; mandic@med.uni-marburg.de

**Keywords:** cottontail rabbit papillomavirus, CRPV, rabbit, head and neck cancer, head and neck squamous cell carcinoma, HNSCC, tumor model

## Abstract

Having suitable animal models is crucial to mimic human disease states and for the successful transfer of experimental data into clinical practice. In the field of papillomavirus research, the domestic rabbit (*Oryctolagus cuniculus*) has served as an indispensable model organism for almost 100 years. The identification and characterization of the first papillomaviruses in rabbits, their carcinogenic potential and their immunogenicity have contributed significantly to the state of knowledge on the genetics and life cycle of papillomaviruses in general, as well as the development of antiviral strategies such as vaccination procedures. Due to the high species specificity of papillomaviruses, only rabbit papillomaviruses (RPVs) can be used for animal studies on papilloma-based tumor diseases in the rabbit. The major focus of this article is on cottontail rabbit papillomavirus (CRPV)-related rabbit squamous cell carcinoma (RSCC). A brief history outlines the discovery and generation of experimentally used RSCC tumors. A comprehensive overview of the current CRPV-associated VX2 carcinoma-based tumor models with a major focus on human head and neck squamous cell carcinoma (HNSCC) tumor models is provided, and their strengths in terms of transferability to human HNSCC are discussed.

## 1. Introduction

Papillomaviruses (PVs) consist of small, nonenveloped, icosahedral DNA viruses [1], which can infect a wide range of species from fish to reptiles, birds and vertebrates [2], as well as humans. The vast majority of the genus *Papillomaviridae* are specific to certain animal species [3], and most strains show high specificity for their respective host. According to the general criteria established by the ICTV (International Committee on Taxonomy of Viruses; https://ictv.global (accessed on 18 July 2024)), the family *Papillomaviridae* has two subfamilies consisting of 53 genera, with 52 genera belonging to the subfamily *Firstpapillomavirinae* and only 1 genus to the subfamily *Secondpapillomavirinae* (ICTV 2022 Master Species List (MSL38, Version 08.04.2023)). With the help of modern molecular biological methods such as next-generation DNA sequencing, new genotypes of the 133 currently known species are still being discovered and characterized.

Human papillomavirus (HPV) strains frequently cause subclinical infections or form benign papillomata that are typically cleared within a period of 2 years. However, a growing number of HPV strains are being identified that are associated with the main types of human cancers. Based on their oncogenic potential, described by the HPV Information Center (an amalgamation of the International Agency for Research on Cancer (IARC) and the Catalan Institute of Oncology (ICO); https://hpvcentre.net (accessed on 26 July 2024)), known HPV strains can be classified into HPV types with a low risk of developing cancer (low-risk types) and those with an increased or higher risk (high-risk types) [4]. The low-risk HPV types 6 and 11 are a frequent cause of benign papillomatosis, while HPV type 16 and HPV type 18 are classified as high-risk HPV types. These high-risk HPV strains are associated with the development of cervical cancer and other anogenital cancers, which are among the leading causes of mortality and morbidity worldwide [5]. Research on the causal involvement of HPV infections in the development of head and neck squamous cell carcinoma (HNSCC), particularly of the oropharynx [6], has promoted the understanding of underlying pathomechanisms, as well as the identification of new therapeutic targets.

## 2. History of CRPV Animal Models

In all susceptible animal species, including humans, infection with PV leads to papillomatosis, which is characterized by the growth of papillomas (warts) that is related to the high tropism of these viruses for squamous epithelia of the skin and mucosa. In the end, it was the formation of warts that subsequently led to the discovery of the first PV as the causative and infectious agent of this skin disease. Historically, knowledge about the infectivity of warts is based on observations by McFadyean and Hobda and Cadeac, as well as Ciuffo, around the turn of the 1900s, who described the transmissibility of warts in dogs [7], horses [8] and humans [9] through cell-free extracts without knowing the actual infectious agent at the time. It was not until systematic studies of papillomas in Eastern Cottontail rabbits by Richard Edwin Shope, who described the etiology of the wart like rank growth and identified a filterable infectious agent as the cause for transmissible cutaneous papillomatosis in Cottontail rabbits (*Sylvilagus floridanus*), as well as domestic rabbits (*Oryctolagus cuniculus*) [10]. Soon after, the cottontail rabbit papillomavirus (CRPV) was identified as the causative agent of transmissible tumor-like diseases and named as Shope virus in recognition of the work of Richard Edwin Shope [11]. The demonstration that infection of epithelial cells with the Shope virus can in some cases cause malignant neoplasms that histologically resemble squamous cell carcinoma represents the first evidence of an oncogenic virus in mammals [12,13,14], following Rous’s description of the oncogenicity of the retrovirus from the genus *Alpharetroviridae*, known as the Rous sarcoma virus, in birds [15,16].

The significance of the Shope virus rabbit model for experimental studies on tumor development was recognized, and efforts to develop a standardized tumor model were intensified. After passaging CRPV-induced lesions from one rabbit to another, Andrewes and Shope observed a change in the type of lesion induced by the fibroma virus at passage 18. Instead of the expected fibromatous lesions as observed in the earlier passages, the lesions now were predominantly inflammatory in character and partly composed of lymphocytes and large mononuclear cells in place of fibroblast-like cells. The investigators suspected this was due to changes in the virus, which they labeled a mutation [17,18,19], and it was precisely this wart model in rabbits that Rous and colleagues used to demonstrate the carcinogenic potential of rabbit papillomaviruses in cottontail and domestic rabbits [13,20,21]. The progression of Shope virus-induced benign lesions into metastasizing carcinomas was observed in up to 5% of infected cottontail rabbits and in up to 75% of infected domestic rabbits [12,22]. These observations laid the foundation for the CRPV rabbit model.

A few years later, another PV strain was detected in small papillomas located on the underside of the tongue in domestic rabbits of the New York metropolitan area [23]. These oral papillomata showed no tendency to develop into cancer but remained benign and often persisted for several months, sometimes for a year or longer, resembling virus-induced skin warts in humans. Again, a previously unknown virus was isolated from these papillomas and used to reproduce the lesions in both domestic rabbits and various wild hare species [23]. This virus, known as rabbit oral papillomavirus (ROPV), exhibited selective specificity for mucous membranes and thus differed clearly from the Shope virus, which causes skin papilloma in rabbits but has proven to be harmless to the oral mucosa. Similarly, as observed for the Shope virus, the immune system also develops immunological resistance to ROPV but without exhibiting cross-resistance. It should be emphasized that these systematic studies and the conclusions drawn have helped to better understand the PV-related transformation into neoplastic cells. They took place in a scientific period in which targeted interventions in the genome were not yet possible and for which the term “pre-genetic modification area” was coined by Cladel and co-workers [24].

## 3. Genome of the Rabbit Papillomaviruses

The first detection of virus particles in skin papilloma was reported by Strauss and colleagues using electron microscopy techniques [25,26], and the advent of molecular biology techniques in the 1970s enabled the cloning of viral DNA. The first reports on the double-stranded circular DNA of the Shope virus appeared in 1963/64 [27,28], followed by the first characterization of HPV DNA in 1965 [29]. Within the next decade, the remarkable plethora of PV types within a species became obvious [30,31,32,33] (for review, see [34]). However, the complete nucleotide sequence of the Shope virus genome was not reported until 1985 by Giri, Danos and Yaniv [35]. CRPV consists of a double-stranded DNA (dsDNA) with a size of 7868 bp. CRPV genes can be divided into two main categories consisting of nine early (E) and two late (L) genes. The two genes of the (L) category code for the major capsid protein (L1), which forms the majority of the capsid that surrounds and protects the viral genome and an additional minor capsid protein (L2) that plays a role in the packaging of the viral DNA and penetration into the host cell. The genes of the (E) category consist of *E1*, *E2*, *E4*, *E5*, *LE6*, *SE6*, *E7*, *E8^E2* (formerly *E9^E2C*) and *E10* (formerly *E8*). *E1* and *E2* are involved in replication and gene regulation and are required for infection. *LE6* and *E7* have a major role in cellular transformation. *E10* and *SE6* promote tumor growth, and *E4* is involved in DNA synthesis and expression of L1 capsid proteins. *E8^E2* represses transcription of the infected host cells, whereas *E5* exhibits a mutation in its start codon and does not appear to be functional [36]. Sequence comparisons with the genome of human PV revealed a high homology to *HPV1a* [37] and *HPV6b* [38], which are associated with deep palmoplantar and genital warts, respectively. In particular, studies on the viral oncogenes *E6* and *E7*, which also play a central role in HPV diseases, characterize their impact in cellular transformation and tumorigenesis.

The identification of the exact PV genome, individual viral genes and the effect of mutations of individual genes have contributed significantly to the fundamental understanding of papillomavirus biology [39], the transmission and replication of PVs in humans and animals [40] and the cause of the oncogenicity of individual HPV strains [41,42]. Knowledge of the interaction of PV with the host immune system has led to the development of antiviral therapies [43], including the development of HPV vaccines [44,45] (for review, see [46]). These vaccines are now an essential tool for the prevention of cervical cancer and other HPV-related diseases in both female and male humans.

Rabbit papilloma and carcinomas are similar in many aspects to human HPV-induced tumors, especially in the head and neck region, making rabbits a highly valuable model for studying the pathogenesis and therapeutic approaches for HPV-associated cancers in humans [47]. In particular, studies on the viral oncogenes *E6* and *E7*, which also play a fundamental role in HPV diseases, characterize their impact on cellular transformation. Interestingly, while the CRPV oncoproteins E6 and E7, as observed for HPV, are required for oncogenic transformation of the cell, there are some differences between these CRPV and HPV oncoproteins. As known for HPV E7, CRPV E7 also inhibits the retinoblastoma (RB) protein [48]. However, in sharp contrast to HPV, CRPV E6 does not bind p53 and also cannot induce its degradation [49]. Instead, Du et al. demonstrated the binding of CRPV E6 (LE6 = long E6 and SE6 = short E6) to another tumor suppressor, hDlg/SAP97, which could explain its oncogenic potential analogous to p53 inhibition [49].

Two classes of RPVs are currently recognized in rabbits: the ROPV and the CRPV. The nomenclature of these two independent virus species is often used unclearly and inconsistently in the literature and should be based on the current International Committee on Taxonomy of Viruses (ICTV) classification. A detailed publication on the origin and current nomenclature was provided by Bernard and colleagues [50].

RPVs show a strict species specificity and strong tropism to epithelial cells. Additionally, both RPVs, CRPV and ROPV, have different primary host specificities within various rabbit populations. They cause different types of papillomata and have different risks for tumor development. ROPV mainly infects domestic rabbits (*Oryctolagus cuniculus*). It is a mucosa-tropic virus that causes papillomatous lesions primarily in the oral cavity, with a major focus on the tongue and lips. The oncogenic potential to cause cancer is rather low, as these papillomata usually remain benign. CRPV predominantly infects wild rabbit species such as the cottontail rabbits (*Sylvilagus* spp.) but can also infect domestic rabbits. This cutaneous-tropic virus causes skin papilloma, especially on the head and neck. It has a significantly higher oncogenic potential, since these skin papillomata can spontaneously progress to squamous cell carcinomas. The host specificity of the two virus species for domestic rabbits makes them ideal candidates for controlled animal experiments. In sharp contrast to CRPV, ROPV has not been unequivocally linked to malignant transformation. However, experimentally, it was demonstrated that ROPV E6, E7 and E8 can act as oncogenes, since cells transfected with these viral genes lead to tumors in athymic mice [51].

Today, NZW (New Zealand White) rabbit animal models exist for both RPVs: the ROPV and the CRPV model. On the other hand, the high tissue specificity represents a major obstacle in setting up and maintaining a standardized CRPV or ROPV rabbit model. The complex life cycles of papillomaviruses, including CRPV, require the complete process of epithelial cell differentiation to complete their life cycle and produce virions, which makes the generation of infectious virus stocks by in vitro cultivation a challenge.

## 4. Induction of RPV-Based Carcinomas

Today, various methods and models have been developed for the induction of RPV-induced papilloma or carcinomas in rabbits with different genetic backgrounds (inbred, outbred, transgenic and gene knockout). Depending on the infectious materials used (viruses isolated from carcinogenic tissue, viral isolates from cloned plasmids and neoplastic CRPV-infected cells/tissue fragments), these models are valuable tools for studying the mechanisms of viral infection, tumor development and the efficacy of therapeutic approaches.

For direct virus inoculation, virus particles are purified from rabbit papilloma and experimentally applied to the epidermis or mucous membranes or injected into the subcutis [52,53]. In the topical application of purified virus particles, the skin, usually on the back, is first lightly scratched to injure the epidermis, and then, the virus suspension is applied to the injured area. This method mimics natural transmission, which can be reproduced by skin lesions and leads to the development of papilloma and possibly carcinomas. However, the collection and production of CRPV stocks is challenging. CRPV can be isolated and purified from the resulting rabbit papilloma but yields only a limited amount of virus stock that is usually sufficient for only a limited number of experiments. Larger virus stocks can be obtained from cell cultures, but the complex life cycles of PV, which are closely linked to the differentiation of host cells, especially epithelial cells, require specialized organotypic (three-dimensional) raft culture systems, in which multiple epithelial cell layers are formed that mimic the natural structure of the tissue and thus enable virus replication, which allows precise characterization of the complete viral life cycle [54,55].

An alternative method is the use of isolated viral CRPV DNA, which is generated by cloning the entire [56] CRPV genome into a vector [24,57,58,59]. This is a reliable molecular option for setting reproducible squamous cell carcinomas of the skin. It is either applied onto scarified rabbit skin or delivered by a gene gun [57,60]. The possibility of testing targeted CRPV gene mutations makes this method particularly suitable for functional studies of individual CRPV genes and for questions concerning the interaction of the virus with the host’s immune system, such as the formation of virus-neutralizing antibodies (VNAs).

Most frequently, cell lines derived from CRPV-induced squamous cell carcinomas are used in the experimental settings. The generation of VX tumor cell lines originated from the work of Smith et al., who sought a method to experimentally reproduce single defined carcinomas to characterize suspected Shope virus variants and associate them with the individual pathological phenotype seen in the tumorigenic papilloma [61]. They are based on Shope virus-induced adult rabbit passable tumors, which were named V1 and V2 in early years. But the nomenclature of the tumor lines was renamed VX in the 1940s, when the first ballistic missile in military history developed during World War II became known under the name “V2”. The tumor lines VX3–VX6 were generated from the VX2 tumor line. All these lines were ultimately not further passaged in the long term. They either lost transferability in an early passage (VX3), were highly ulcerative and could only be transplanted into rabbit pups (VX4), cells died very quickly in vivo (VX5) or were simply not pursued further (VX6) [61]. In addition, Georges and coworkers used the VX2 tumor to generate two cell lines, VX2T and VX2R, and noticed a loss of transplantability, especially for the VX2R cell line [62]. However, only the original VX2 tumor line and another VX7 tumor line, derived from newborn rabbits [63], could be preserved until today by serial transplantation into adult domestic rabbits.

Despite several early contradictory results, it was concluded that the CRPV genome in VX cell lines is integrated into the host chromosome, resulting in a unique expression pattern in the VX2 and VX7 tumor cells [64,65,66,67,68]. Complete virus replication is not found, or only to a small extent, in the rabbit warts and is absent in the carcinomas [22]. The lack of expression of late viral genes and thus complete viral replication leads to the loss of VNA formation, which was the case in the early transplantation experiments of warts and tumors in cottontail rabbits [69,70,71,72,73].

In this regard, the VX7 tumor line is an exception. VX7 tumors still produce virus-neutralizing antibodies even after more than two decades of serial transplantation and thus show a strong interaction with the adaptive immune system of the host. In contrast, VX2 carcinoma-derived tumor cells have lost their ability to produce viral antigens and induce VNAs [74] during serial passages, but transcription of the viral genome in the cells is sufficient to maintain malignancy. This difference in immunogenicity predestines the two tumor cell lines for different questions regarding vaccine development, tumor angiogenesis and metastasis in the course of advanced tumor growth without effective immune surveillance but also the development of new anticancer therapies.

## 5. Rabbit as a Model Organism

Rodents are most commonly used in biomedical research, because they have a short generation time, are easy to handle and breed and can be genetically manipulated. The fact that RPV tumors grow and metastasize in various parts of the body has led to the establishment of a variety of animal models used in specialized research areas, particularly in radiology, toxicology, surgery, virology and oncology. The rabbit VX2 tumor model is of central importance for the development of new therapeutic approaches in the field of otorhinolaryngology—head and neck surgery and oral and maxillofacial surgery—particularly due to the possibility of experimental tumor seeding in areas of the body that are difficult to access surgically, such as the head and neck.

Among the different rabbit strains, NZW rabbits are almost exclusively used for research purposes, as this strain shows robust health and low aggressiveness [75]. Compared to larger animals such as dogs, pigs or monkeys, rabbits are cheaper and ethically easier to handle and care for. Especially in the fields of radiology, surgery and oncology, the advantages of NZW rabbits are their size, robustness and immunocompetent status. Their size is large enough to carry out an anatomical and tumor-spatially finely graded evaluation of surgical or radio(chemo)therapeutic interventions but also small enough to be accommodated in most laboratory facilities. An important prerequisite when using rabbits as a model organism for viral research is their natural susceptibility to RPVs. They are a natural host, in which the RPV undergoes natural infection and disease development, making the studies more realistic and relevant. In addition, it also means that the animals used in such trials are allowed to have a normal immune status. This offers an invaluable advantage for immunological questions regarding virus-driven tumor development, as well as lymphogenic spread to local lymph nodes and distant metastases to the lungs. The rabbit models enable controlled experiments in which the infection dose, infection site and other experimental conditions can be precisely controlled. Today, there are various comprehensively characterized and established CRPV rabbit models, especially the VX2 carcinoma model that can be used to reliably produce papilloma and squamous cell carcinomas that show consistent and reproducible tumor development in a variety of tissues and functional systems (Figure 1). This is highly important for the systematic investigation of disease mechanisms and, eventually, the efficacy of therapeutic approaches.

## 6. Rabbit VX2 Tumor Models in the Head and Neck Area

### 6.1. Head and Neck

Animal models for tumors in the head and neck region must reflect the anatomical localization of the primary tumor and its clinical course in humans in order to test new diagnostic, surgical, radiological and alternative methods for use in tumor therapy. HNSCC is a commonly detected cancer with approximately 760,000 new cases diagnosed annually worldwide [76]. About 25% of HNSCC—in particular, those located in the oropharynx—are (human) papillomavirus (HPV)-associated [77], similar to the VX2 carcinoma of rabbits. Barsouk et al. pointed out a growing prevalence of HPV-associated HNSCC, predicting that HPV will surpass tobacco as a risk factor for HNSCC [78]. Therefore, papillomavirus-driven tumor models, such as the CRPV-associated VX2 carcinoma, may become increasingly more significant to experimentally address new therapeutic approaches prior to use in HNSCC patients.

The grafted model tumors must have the ability to form solid tumors in different regions in the head and neck area, which, according to the clinical situation, also include regions that are difficult to access surgically, which is, in fact, the case for CRPV-based tumor cells in general and VX2 tumor cells in particular (Figure 2). There are several similarities between the CRPV-driven VX2 carcinoma and human HPV+ HNSCC. Clinically, both tumors primarily exhibit a lymphatic metastatic spread to locoregional lymph nodes [79,80], which is one major aspect why the auricular VX2 carcinoma was initially considered a suitable model system for human HNSCC. Histopathologically, both tumors, the CRPV-associated VX2 carcinoma and HPV+ human HNSCCs, exhibit a low differentiation level (basaloid) that typically is seen in PV-driven tumors. This is also highlighted by the reduced keratinization level seen in Figure 3. Furthermore, in our previous studies, we demonstrated tumor-infiltrating lymphocytes (TILs) to correlate with the therapy response of VX2 carcinomas [81], which is in accordance with observations in human HPV+ HNSCC [82]. Hereinafter, various VX2 rabbit models are discussed according to the localization of the primary tumor and their transferability to the human situation.

#### 6.1.1. Ear

Since the ear is a peripheral structure, far from important organs, vessels and nerves, tumor growth and experimental manipulations cause less discomfort to the animals. Additionally, auricular tumors are easily accessible for observation, measurements and manipulation. The NZW rabbit VX2 carcinoma of the ear represents a compatible model system to human HNSCC that can be used for studies on angiogenesis, tumor growth and metastasis of head and neck squamous cell carcinomas, thereby helping to evaluate the efficacy and risks of radiological, surgical, chemo- and immunotherapies of HPV-based HNSCC. The VX2 ear model is based on the transplantation of VX2 tissue pieces or usually the injection of VX2 tumor cells into the subcutis of the ear, which leads to the growth of a solid tumor within 2 weeks [83,84]. The growth is associated with early angiogenesis in the local area of the tumor [83,84] and a restructuring of the micro-angioarchitecture of tumor-supplying vessels [85], as observed in HNSCC tumors [86,87].

Another key element of this animal model is the rapid metastasis to the local lymph nodes [80] and, subsequently, to the lungs (Figure 2). In the auricular primary VX2 tumor, like in HNSCC [88], various matrix metalloproteases (MMPs) are overexpressed [89,90], which are essential for tumor infiltration into the surrounding tissue and metastasis [91]. The exact topography of the cervicofacial lymph nodes has been characterized [92] and served as the basis for the evaluation of the lymphogenic metastatic spread of auricular VX2 carcinoma. The histological pictures of cervical and facial lymph nodes of NZW rabbits are comparable to those observed in humans. Both show similar characteristics regarding the location and type of lymphatic follicles. The location and surgical accessibility of the caudal and rostral mandibular, as well as defined parotid lymph nodes, enabled further detailed investigations regarding the lymphogenic metastatic spread of induced VX2 carcinomas [80]. Knowledge of the lymph node topography and the time frame of VX2 metastasis made this VX2 ear model suitable for sentinel node biopsy studies in HNSCC [93]. These examinations could be used to diagnose the success of surgical, radiological and drug treatment. To date, a large number of treatments have been tested and evaluated with the NZW VX2 carcinoma ear model.

**Figure 3 viruses-16-01722-f003:**
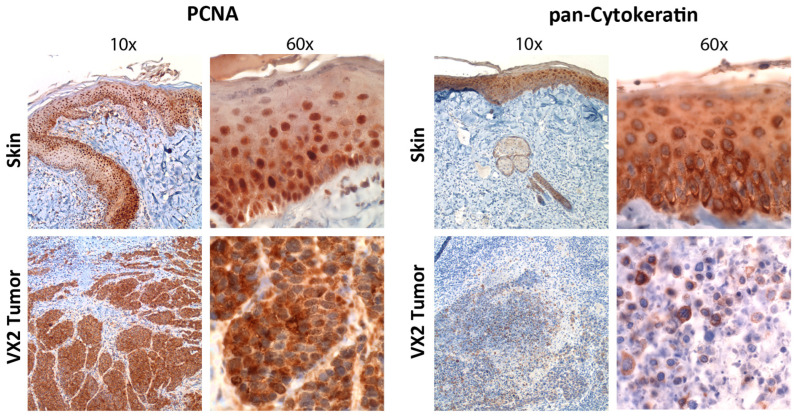
Immunohistochemistry of formalin-fixed paraffin-embedded (FFPE) rabbit normal skin and VX2 tumor tissue. Low (10×) and high (60×) magnification images showing immunoreactivity of normal rabbit skin and the VX2 carcinoma to an antibody directed against proliferating cell nuclear antigen (PCNA; cat#: sc-56, Santa Cruz Biotechnology, Inc., Dallas, TX, USA) that plays a crucial role in DNA replication and repair, as well as cell cycle regulation. Note the strong nuclear staining in the basal layer of the normal epidermis, which harbors proliferating and stem cells and the abundant expression in the nucleus but also the cytoplasm of nearly all tumor cells, pointing to the high proliferative capacity of the VX2 cells. Only minimal immunoreactivity is seen in cells of the neighboring connective tissue. Major immunoreactivity to an antibody directed against pan-cytokeratin (cat#: sc-8018, Santa Cruz Biotechnology, Inc.) was observed in the epidermal layer of normal skin, whereas only minor cytokeratin positivity is seen in the VX2 tumor, which is, in accordance with the fact that VX2 tumors, as well as other papillomavirus-driven cancers, exhibit a low grade of cellular differentiation. Immunohistochemistry was performed as previously reported [94]. The animal experiments from which the VX2 tumor tissue samples were collected were approved by the Regional Council of Giessen, Germany (V54- 19c20-15(1) MR, No. 34/2011), in accordance with the German Animal Welfare Act.

A surgical approach investigated the feasibility of resection of the auricular VX2 carcinoma, its influence on regional and distant metastases, the surgical method of resection and the general prognosis [95]. The VX2 auricular squamous cell carcinoma initially spreads along the lymphatic vessels and involves the well-described cascade of draining lymph node stations. In later stages, the metastatic process also spreads to distant areas, especially the lungs. Local recurrences of VX2 carcinomas have been observed despite extensive resection in sano with a large margin, most likely due to microembolization [95]. Surgical en bloc resection with cold steel and surgical en bloc laser resection carry a similar risk of postoperative metastases [96], whereas the piecemeal surgical laser resection method has a potentially increased rate of regional lymphatic metastases [97]. Metastatic spread of tumor cells cannot be excluded with certainty by any of these methods. This chronologically and topographically reproducible process is analogous to the process of regional and distant metastatic spread of human HNSCC [98] and corresponds to the clinical situation with resected primary tumors and the wait-and-see approach in the neck.

Embolization of tumor-supplying vessels is a minimally invasive alternative treatment strategy for a wide range of tumor entities. For the auricular VX2 carcinoma, this method has been experimentally established using dextran microspheres [99] and holmium-poly(L-lactic acid) microspheres [100,101]. Such microspheres can serve as carriers for the targeted delivery of various active drugs that are released locally in the tumor area during embolization. There are no other relevant studies on the auricular VX2 rabbit model for the microspheres mentioned above. Dextran microspheres are used in different VX2 rabbit models as carriers for substances in the context of trans-arterial chemoembolization (TACE), in particular in the liver of different VX2 models [102]. For the auricular VX2 rabbit model, as well as all other anatomical sites of VX2-induced HNSCCs, there is currently one publication about the deployment of TACE using doxorubicin-loaded chitosan microspheres, focusing on the synthesis and loading of the microspheres with the chemotherapeutic agent [103]. Intra-arterial chemotherapy is increasingly being used as an alternative treatment method for advanced-stage HNSCC [104], thus categorizing VX2 models in the head and neck region as suitable for testing targeted delivery strategies in difficult-to-access HNSCC.

Further innovative cancer treatment strategies for inoperable HNSCC are immunotherapy approaches [105,106], which focus on tumor destruction based on the body’s own immune system (for a description of the modes of action, see National Cancer Institute, https://www.cancer.gov/about-cancer/treatment/types (accessed on 16 August 2024)). One way of triggering or enhancing an anti-tumor reaction is the targeted administration of immunomodulating drugs such as pro-inflammatory cytokines or checkpoint inhibitors such as programmed cell death protein-1 (PD-1), programmed cell death-ligand 1 (PD-L1) and cytotoxic T-lymphocyte associated protein 4 (CTLA-4). Such immunomodulators can be either administered intravenously or packaged in microsphere carrier systems and introduced into the tumor area via the tumor-bearing vessel. Today, there are commercially available monoclonal antibodies that functionally block rabbit PD-L1 and LAG-3 signaling pathways [107] (Roy et al. 2019) or can be used for immunohistological detection of PD-1, PD-L1 and CTLA-4 in rabbits [108].

It is crucial for the success of immunotherapy that therapeutically induced tumor destruction does not lead to a complete disintegration of the tumor mass and thus to the loss of immunogenic tumor-specific epitopes but instead enables the release of native tumor-specific antigens. Modern interventional radiotherapies such as radiofrequency ablation (RFA), cryoablation or photodynamic therapies (PDT) enable such “epitope-preserving” tumor destruction. The combination of interventional radiotherapies and immune-supporting adjuvant therapies is one of the most promising alternative treatment approaches for numerous tumor entities today. Also, experimental approaches currently available for VX2 tumors in the head and neck region include cytokine-mediated immune stimulation, TACE, magnetic drug targeting, RFA, PDT [104] and the unconventional approach of an oxidative therapy by using an O_3_/O_2_-gas mixture [109].

Deploying the biauricular VX2 rabbit model that is characterized by the installation of one VX2 tumor in each ear, an immunological approach in which interleukin-2 (IL-2) was applied peritumorally around a solid tumor showed high remission rates of the tumors, which also included the contralateral tumors not treated locally with IL-2, as well as existing lymph node metastases [109,110]. The involvement of the immune system in tumor remission was additionally demonstrated by rechallenging experiments on cured animals, as these had developed protection against the re-growth of the tumor. For a more targeted delivery of different anticancer drugs into the tumor, dextran microspheres were successfully developed as carriers on the biauricular VX2 rabbit model. These microspheres enable transient local embolization in the tumor area by TACE without leakage of the microspheres into the blood and thus represent a model system for testing new active substances in the head and neck area [99]. Another microparticle formulation designed for targeted delivery of a chemotherapeutic agent developed on this VX2 model consists of a poly vinyl pyrridone/poly vinyl alcohol (PVP/PVA)-based magnetite hydrogel loaded with the chemotherapeutic agent bleomycin A5. These ferrofluid particles were enriched in the tumor area with the help of a magnet, transiently embolized the blood flow and released bleomycin A5 [111]. A therapy-based reduction in auricular tumor size was demonstrated. Intra-arterial embolization with radioactive microspheres (TARE) consisting of radioactive holmium-166 poly(L-lactic acid) microspheres resulted in a high rate of VX2 tumors in complete remission [101]. The therapeutic success in this experimental animal model supports the use of TARE in HNSCC in the same manner as it was used for the treatment of other malignancies, such as hepatocellular carcinoma (HCC) [112].

The therapeutic efficacy of a combination therapy of RFA and immunotherapy was demonstrated in the auricular VX2 tumor model. For this purpose, a twin VX2 tumor model was used, in which primary VX2 tumors were generated simultaneously in the lung and ear. Only the lung tumor was treated by a combination of RFA and immune stimulation, which was done by intra-tumoral injection of the immunostimulant OK-432. The remote effect of this combination therapy was observed in the untreated VX2 tumor located in the ear. Increased numbers of inflammatory cells were found in the auricular tumors after combination therapy and the tumors regressed [113]. Findings from these focal interventional ablation procedures may provide the basis for the development of effective systemic anti-tumor therapies [114].

For photodynamic therapy (PDT) approaches, there are currently two studies on PDT on the auricular VX2 model. Using Photofrin II as a photosensitizer, non-invasive real-time information on the influence of the light dose on the oxygen supply in the tissue during irradiation was obtained. The data show the feasibility of inducing irreversible tissue hypoxia in the tumor as a function of irradiation [115]. A very well-founded effect of PDT to provoke an effective anti-tumor immune response was shown by Pang and colleagues [116]. Using the biauricular VX2 rabbit model, they show effective local cancer cell apoptosis in the irradiated ear and distant tumor inhibition in the contralateral ear by deploying Prussian blue nanoparticles (PB NPs) as photosensitizers [116]. Flow cytometric analyses demonstrate significantly increased numbers of tumor-killing CD3+ CD8+ CTL in the spleen and in the contralateral distant tumor, as well as systemically increased cytokine levels of TNF-α, INFγ and IL-10 in the serum measured by ELISA. The data thus confirm the feasibility and effectiveness of PDT as an immune-based therapy for HNSCC.

However, the induction of an anti-tumor immune response does not necessarily require a local intervention on the tumor tissue but can probably also be induced by a systemic stimulus away from the tumor. In case of the auricular VX2 tumor model, a more pathophysiological experimental approach led to an effective immune response directed against the VX2 tumor. This therapeutic approach is based on a repetitive oxidative stimulus in the peritoneal cavity induced by insufflation of a medical O_3_/O_2_ gas mixture. This stimulus lead to an abscopal effect characterized by a systemic upregulation of leukocytes in the blood [109] and a massive local infiltration of CD3+ T cells into the distant auricular VX2 tumor. A complex immune reaction takes place in the tumor stroma, which involves upregulation of various pattern recognition receptors, signal transduction factors, costimulatory molecules involved in antigen presentation and cytokines/chemokines [81]. The tumors remit and reimplantation experiments in healed animals demonstrate acquired protection against the tumor, which was observed in the IL-2 study, the PDT approach, as well as the combination therapies of RFA/OK-432 and PVP/PVA/bleomycin mentioned above. The mechanism of action that leads to this abscopal effect is not yet understood in any way. In case of the O_3_/O_2_-PP, a hypothetical explanation might be that the powerful oxidative stress within the peritoneal cavity might kill potentially immunosuppressive cells, including myeloid-derived suppressor cells (MDSCs) and Treg cells, which, so far, have inhibited a tumoricidal immune response. Consequently, leukocytes might become activated directly or via the generation of ROS (reactive oxygen species) and LOP (lipid oxidation products). Under this condition the population of TILs might preferentially consist of activated TILs, but lack suppressive MDSCs and Treg cells [117]. However, the O_3_/O_2_-PP model is suitable for the identification of potential diagnostic, prognostic and therapeutic biomarkers for successful HNSCC therapy. In tumors that were in remission due to O_3_/O_2_-PP treatment, the ERBB gene profile was identified as a predictor of treatment response [94].

#### 6.1.2. Paranasal Sinuses and Nasal Cavity

In one of the first scientific studies on the role of fibrinolysis in cancer metastasis, the maxillary sinus VX2 model was deployed in 1973 [118]. It was shown that experimentally induced fibrinolysis leads to a high increase in the number of VX2 distant metastases. The characterization of fibrin and fibrinogen today is a central field in oncology with high therapeutic potency [119,120]. A second study in 1982 used the maxillary sinus VX2 model to characterize the expression levels of superoxide dismutase 2 (SOD2) in tumor cells and in various other organs of tumor-bearing rabbits. SOD2 plays an important role in protecting cells from oxidative stress. The reduction of radical oxygens contributes to the survival, progression and metastasis of tumors, among other things [121,122]. Interestingly, the data showed that VX2 tumor cells alone showed no SOD activity and a decrease in SOD activity during the course of the tumor disease was only observed in the liver [123].

#### 6.1.3. Oro-Maxillofacial Region

Oral cavity squamous cell carcinoma (OCSCC) is the most prevalent form of head and neck cancer with nearly 390,000 new cases diagnosed globally each year [124]. Most OCSCCs arise from the oral tongue, lips and floor of mouth [125] but can affect the maxillofacial region. Due to the anatomical similarities between rabbits and humans, the experimental implantation of a VX2 tumor can mimic the clinical situation of different tumor localizations in the oro-maxillofacial region. Experimentally, VX2 tumors can be grafted in the buccinator muscle, the mucosa of the oral cavity, the tongue or the floor of mouth.

VX2 tumors in the buccinator muscle were used to test a radiotherapeutic approach in combination with antisense therapy, surgery as well as a PDT. The focus of the radiotherapeutic approach using this model was to reduce the frequently occurring hypoxic effects during radiotherapy on solid tumors, as hypoxia increases the expression of vascular endothelial growth factor (VEGF) in tumors, which then can promote tumor growth by increasing angiogenesis. Blocking VEGF formation during radiotherapy is a rational therapeutic approach [126], which was examined by a direct intra-tumoral injection of VEGF antisense oligonucleotides during radiotherapy of buccal VX2 tumors. Knock down of VEGF expression showed an inhibitory effect on angiogenesis [127]. Significant hypoxia induction by irradiation of this buccal VX2 tumor was confirmed in a later study using semi-quantitative dynamic contrast-enhanced magnetic resonance imaging (DCE-MRI) [128]. In view of the poor prognosis of head and neck tumors due to their invasiveness and lymphatic metastasis, a newly developed nanoprobe was tested on the buccinator muscle VX2 model for its suitability of assessing tumor margins during surgical resection [129]. The aim during surgery is to avoid unnecessary resection of neural structures. This nanoprobe enables MRI-based preoperative imaging of the primary tumor, as well as existing lymph node metastases and an improved visualization of the tumor margins during the surgical procedure, which is realized by multimodal surface-enhanced resonance Raman spectroscopy (SERRS) in real time. Surgical treatment of maxillofacial VX2 tumors located in the buccinator muscle using this AuS-based multimodal MR/SERRS probe was more effective than conventional white light-guided surgery and prolonged the survival of the rabbits [129]. A near real-time image-guided method for sentinel lymph node biopsy (SLNB) was also tested on lymph node metastases derived from VX2 tumors located in the masseteric muscle. Based on the fluorescent dye indocyanine green (ICG), cone beam CT (CBCT) and fluorescence imaging in the near infrared range (NIR) were used to generate images and 3D reconstruction for the localization of existing lymph node metastases during the SLNB procedure [130]. For non-invasive ablation using the VX2 buccinator model, a PORPHYSOME (PS) nanoparticle formulation was developed and tested to enable effective PDT. The PS nanoparticle showed good tumor selectivity and achieved complete ablation in 100% of tumors after repeated superficial and interstitial PDT [131].

#### 6.1.4. Pharynx and Larynx

For the investigation or treatment of tumors of the larynx or nasopharynx, rabbit models were established in which a VX2 tumor was installed either on the vocal cord [132] or in the nasopharyngeal posterior wall adjacent to the clivus [133]. Histological analyses of these VX2 tumors showed features consistent with human carcinomas, including invasiveness, superficial ulceration, lymphatic spread and airway involvement. To better determine the invasiveness of HNSCC tumors preoperatively and thus reliably define the necessary extent of surgical resection, the diagnostic reliability of non-invasive high-resolution optical coherence tomography (OCT) in combination with endolaryngeal ultrasonography (EUS) was tested with VX2 tumors located on the vocal cord [134]. The VX2 tumor in the nasopharynx was also used as a model system for the diagnostic evaluation of ^18^F-Fluorodeoxyglucose (FDG) and ^18^F-Fluorothymidine (FLT) PET/CT [133]. The effect of the invasiveness and the space occupied by the tongue VX2 tumor on possible damage to local nerve branches in the tongue was evaluated by histopathology. It was found that the increase in tumor volume lead to massive degenerative changes in locally branches of the hypoglossal and lingual nerves and may be the trigger of clinical symptoms such as paralysis or tongue movement disorders, which occasionally occur in some patients with lingual carcinomas [135].

For the staging of HNSCC, the differentiation of metastatic and inflammatory lymph nodes is of central importance. For this, imaging of vascular structures in the regional lymph nodes is an essential parameter [136], but the reliability of color Doppler ultrasonography and the value of nodal vascularization patterns for differentiation of benign and malignant lymphadenopathies must be demonstrated. For this purpose, cervical lymph node metastases in the head and neck region were induced by implanting a VX-2 tumor on the floor of the mouth, followed by monitoring its vascular development over time using color Doppler ultrasonography [137]. In metastatic lymph nodes, changes in blood flow measured by color Doppler sonography initially showed hypervascularization and the appearance of avascular areas at later stages, whereas reactive lymph nodes did not show any biphasic changes. The image-based data were confirmed by subsequent histopathology and the method was validated for clinical diagnostics.

Like all VX2 tumors in the head and neck region, the tongue VX2 tumor shows lymphogenic metastasis associated with an increase in collecting lymphatic vessels in the tumor area [138]. Local administration of the angiogenesis inhibitor TNP-470 via the ear vein inhibited the growth of these lymphatic vessels, suggesting that the tumor-inhibiting effect of TNP-470 is mediated not only directly via the tumor-supplying vessels themselves [139] but also on the formation of new lymphatic capillaries around the tumor [140], which could explain the reduced number of metastases in the tongue VX2 model [139]. The lymph node metastases created in this VX2 model were used in a series of radiological imaging methods to test the suitability of various contrast agents for visualization and diagnostic suitability [141,142,143,144] and to test procedures for image-guided biopsy of sentinel lymph nodes [145,146]. The relationship between blood vessel density (BVD) and free platinum concentrations during intra-arterial chemotherapy was investigated in the primary VX2 tumors of the tongue itself. In the clinic, BVD is considered a reliable prognostic factor for the response to therapy [147]. The study by Takagi and colleagues confirms this assumption by comparing the data obtained experimentally with the tongue VX2 model with clinical data from patients with squamous cell carcinomas of the oral cavity and oropharynx who had undergone targeted intra-arterial chemoradiotherapy with carboplatin [148]. The efficacy of intra-arterially applied carboplatin in combination with paclitaxel on the VX2 tongue tumor and associated metastases in rabbits was proved by high apoptosis rates of the tumor cells [149].

#### 6.1.5. Thyroid Gland

For the thyroid VX2 rabbit model, there is only one publication to date showing the application of porphyrin-HDL nanoparticle (PLP)-based PDT. The data show a tumor-destructive effect of PDT in the form of significant and specific apoptosis localized in the tumor tissue but not in the surrounding normal tissue, including the trachea and recurrent laryngeal nerve. These experimental data confirm the minimally invasive PDT method as effective and with few side effects [130]. A long-term survival study in humans shows that PLP-PDT enables complete ablation of the tumor tissue and spares thyroid tissue and the recurrent laryngeal nerve [150] and can therefore be classified as a minimally invasive but effective possible alternative to thyroidectomy in the treatment of thyroid cancer.

## 7. Other Rabbit VX2 Tumor Models

### 7.1. CNS

In the central nervous system, the VX2 carcinoma rabbit brain model [151] is primarily used to study neuro-oncological issues. The major focus here is on the imaging of brain tumors [152] or edema [153], establishing of surgical techniques [154], radiation regimes [155] or photodynamic therapies [156]. The intravertebral VX2 spinal cord model [157] is used to study oncological space-occupying lesions for paraparesis [158] and paraplegia [159] and to develop nonvascular interventional therapy techniques [160] and vertebroplasty [161].

### 7.2. Thorax

In the thorax area, various surgical methods for breast cancer are being investigated for feasibility, safety and radiological-diagnostic issues using the NZW rabbit VX2 breast solid tumor model [162,163,164].

#### 7.2.1. Lung

For the VX2 lung model various implantation techniques of VX2 carcinomas exist. In direct application, VX2 cells are implanted into the lungs via catheter [165], bronchoscopy [166,167], surgical opening of the intercostal space [168] or percutaneous puncture injection [169,170].

#### 7.2.2. Pleura

For peripheral induction, VX2 cells are injected into the subcarinal mediastinum [166]. In the rabbit pleura cancer model, VX2 tumor cells are inoculated directly into the pleural cavity [171].

#### 7.2.3. Trachea

The trachea VX2 tumor model was recently developed to investigate bronchoscope interventional treatment approaches for malignant airway stenosis. The intra-tracheal tumor is installed by bronchoscope-assisted implantation of VX2 fragments into the submucosal layer of the airway, resulting in a solid tumor after 2–3 weeks, leading to tracheal stenosis [172]. The clinical goal is to develop new stents, laser and spray cryotherapy in the context of bronchoscope interventions to release malignant airway stenoses [173,174,175].

#### 7.2.4. Esophagus

Depending on the anatomical site of transplantation, the esophagus VX2 model can be used to generate tumors that differ in their invasiveness and space requirements as they grow and mimic human esophageal squamous cell carcinoma (ESCC). Implantation in the thoracic section of the esophagus can be performed surgically or endoscopically [176]. With both methods, as is the case in many other VX2 models, the implantation of small tumor fragments has proven to be more favorable than cell suspension. The surgical method is performed through the abdominal cavity into which the esophagus is briefly pulled down through the esophageal hiatus in order to transplant the VX2 fragment into the esophageal muscle, whereas the endoscopic method used the submucosal layer [177]. The developing VX2 tumors in the muscle are characterized by intraluminal tumor growth and greater invasiveness in neighboring organs compared to endoscopic implanted tumors, which have a higher rate of severe esophageal stricture and of intra-luminal tumor growth. In all cases regional lymph node metastases occur [79,176]. The different pathological characteristics of the two esophageal VX2 models are used for studies on surgical and minimally invasive treatments. These essentially comprise the development of new stents [178,179] and the assessment of new diagnostic imaging procedures [180,181,182].

### 7.3. Abdomen

#### 7.3.1. Urogenital System

The renal VX2 carcinoma model is most frequently used in the urogenital tract. VX2 tumor cell suspensions [183] or tumor fragments [184,185] were transplanted subcapsular within the renal capsule to generate a local tumor. In the VX2 bladder tumor model, VX2 cell suspensions were injected into the lamina mucosa [186] and in the uterine VX2 tumor model, which served as an orthotopic model of endometrial cancer and retroperitoneal lymph node metastasis, the tumor cells are implanted into the lamina muscularis mucosae [187,188].

#### 7.3.2. Gastrointestinal Tract

VX2 rabbit gastrointestinal models include the generation of solid VX2 tumors in the wall of the stomach [189,190], caecum [191], colon [189] and rectum [192]. In addition to radiological, surgical and chemotherapeutic aspects relating to the primary tumors, these models also represent a model of peritoneal carcinomatosis. The same applies to VX2 tumors implanted directly into the peritoneal cavity [193]. The VX2 pancreatic carcinoma model provides information on the spread of tumors within the pancreas [194] and offers experimental approaches for testing interventional therapeutic approaches in pancreatic cancer treatment [195,196].

#### 7.3.3. Liver

The liver VX2 carcinoma model is by far the most frequently used model system. The grafting of the tumor cells into the liver can be carried out by various approaches. After surgical opening of the abdominal cavity, either a freshly obtained VX2 tumor cell suspension (cell suspension method) [197,198,199,200,201] or small VX2 tumor fragments (tissue fragment method) [199,202] can be transplanted into the liver. It has been shown that intrahepatic implantation of solid tumor fragments is more reliable than injection of a cell suspension, as it has a higher “tumor take rate” and a lower incidence of multimodal liver tumors and extrahepatic metastases [199,203,204,205]. To minimize tumor leakage and atopic seeding of VX2 tumor cells, the tumor fragment can be embedded in a gelatin sponge for installation [206]. Another route of application that is less invasive than open laparotomy and reduces the associated risk of infection is percutaneous injection under computed tomography (CT) or ultrasound (US) [205,207,208,209,210] guidance, whereby the embedding of VX2 tumor cells or tumor cell fragments in gelatine [211] also proves to be advantageous in this minimally invasive procedure. To complete all possible installation routes, the perfusion of VX2 tumor cells via the hepatic artery or portal vein must also be mentioned, but this method leads to an uncontrollable distribution of tumor cells in the abdomen and thorax [212]. The excellent review by Florentine Pascale and coworkers provided a comprehensive overview of the rabbit hepatic VX2 tumor model in terms of clinical and imaging features, vascularization, histopathology and highlights the importance of this tumor entity for current immuno-oncological issues [213].

#### 7.3.4. Spleen

The generation of VX2 tumors in the spleen is rather uncommon and is used in the current literature in only one publication as a model for liver metastases [185] and in another publication for the visualization of spleen metastases in radiological diagnostics [214].

#### 7.3.5. Lymph Nodes

Tumors in lymph nodes are almost exclusively due to metastases from primary VX2 tumors in various organs. In the most frequently used method of intramuscular injection of VX2 tumor cells into the hind limb or thigh, lymph node metastases have been described in popliteal [215,216], iliac [217,218] and retroperitoneal [219] lymph nodes. In the case of subcutaneous injection, metastases are found in the regional lymph nodes of the popliteal fossa [220,221], the axilla [222] and in the head and neck region [92,223], depending on the tumor application site. Other rabbit models for the formation of lymph node metastases in the head and neck region are based on metastasis of installed primary tumors in the oral cavity [130] or the tongue [142,143]. Further transplantation sites of the primary tumors are the mammary gland for the generation of metastases in the axillary lymph nodes [224], the uterus with metastasis formation in the retroperitoneum [188] and the rectum for the induction of iliac lymph node metastases [225].

All these VX2 tumor models aim to generate lymphogenic metastatic VX2 primary tumors, which are almost exclusively used in the development of diagnostic imaging techniques. However, metastases only develop at an advanced stage of the disease, so that only limited time remains to address immuno-oncological questions before the animal dies, which to some extent can be prolonged by surgical removal of the primary tumor [226]. Although direct injection of VX2 cells into lymph nodes leads to the formation of a solid tumor within the lymph node and thus certainly prolongs the analysis time of tumor progression, its suitability for immunological questions must be critically discussed [227].

### 7.4. Bone

Since the VX2 carcinoma is not a primary tumor model of the bone, it could serve as a metastatic bone tumor model. The most commonly used injection of fresh VX2 cell suspensions from the muscle is into the tibia to study bone marrow tumor growth [228], regeneration [229] and its effect on metabolism [230,231]. The tibia model mimics human bone marrow tumors for biopsy technique evaluation [232,233], radiological diagnostics [234,235] and various interventional therapies [236,237,238]. Also, the femur [239,240,241] and the iliac crest [242] are deployed in current bone models.

### 7.5. Skin

In addition to the use of subcutaneous VX2 tumors as primary tumors for metastases (see above), the primary tumors can also be investigated directly due to their easy accessibility. The early transplantation experiments by Rous and colleagues in which virus, viral supernatants or VX2 cell suspensions were applied to scratched skin have already been mentioned above. Today, dermal VX2 tumors are transplanted either by surgical implantation of tumor fragments under the skin or by percutaneous injection of fresh VX2 tumor cell suspensions. Current studies on subcutaneous VX2 tumors serve as models for minimally invasive targeted tumor therapies such as neutron capture therapy [243], internal radiation therapy [244,245,246,247], irreversible electroporation (IRE) therapy [248] and microwave-based ablation techniques [185,249].

### 7.6. Muscle

In the vast majority of animal experiments with VX2 carcinoma, the tumor cells are first passed through the muscle of a donor rabbit to increase tumorigenicity. In this case, the VX2 cells usually originate from a frozen VX2 tumor cell stock, which is injected into the muscles of the hind leg either directly after thawing or after a short in vitro cultivation. A normal syringe needle is used for injection. After muscular implantation, few solid tumors with irregular tumor morphology and a pronounced necrotic center develop due to the anatomical features of the fine fibers and the intermuscular spaces. After tumor formation, VX2 tumor material can be isolated for the purpose of generating VX2 tumors at various locations in the body. This type of application is sufficient, as the donor tumor is processed in a targeted manner, and only identified tumor material is fragmented or suspended largely free of immune cells, pus and connective tissue. However, skeletal muscles themselves are also used as target organs for analyses, particularly of chemo- and radiotherapy techniques, due to their free accessibility. Specific questions concerning muscular VX2 tumors require homogeneous growth of only one tumor, which has a spherical shape at best. Here, the dual muscular tumor model on rabbits by Meng and coworkers represents a way to generate precise and stable tumors in two separate skeletal muscles by inserting VX2 tumor strips, which are obtained directly from the donor tumor using a biopsy needle and then pushed into the muscle through a thick needle sheath [250]. This model enables the analysis of possible remote effects of minimally invasive treatments of a tumor on other tumors scattered in the body. In radiology, this phenomenon is known as the abscopal effect [251], which describes an anti-tumor effect of radiotherapy on distant tumors or metastases [252]. The combination of radiotherapy with immunotherapy can enhance the abscopal effect and appears to be an early description of immuno-oncological effects [253].

### 7.7. Blood

The direct injection of VX2 cells into blood can be used to generate metastases in the target area of the respective vessel. Intra-arterial injection of a VX2 tumor cell suspension into the internal carotid artery has been described as a reproducible model for metastatic brain and eye tumors [254]. Injection of VX2 tumor cells into the renal or hepatic and mesenteric arteries generates tumors in the kidney [255] or liver [212], respectively. More frequently, injections into the venous system are reported which, when injected into the ear vein, lead to metastases in the lungs [256,257], and injections into vessels of the portal vein system lead to liver tumors [212]. VX2 models that resemble the clinical picture of vascular stenoses due to tumor space-occupying lesions are available for the superior vena cava syndrome, in which a VX2 tumor cell suspension is injected into the anterior wall of the right superior vena cava. This superior vena cava obstruction VX2 model is used to analyze the respiratory symptom and venous stasis [258]. The VX2 portal vein tumor thrombus (PVTT) model, in which a VX2 tumor fragment is attached within the main portal vein, represents the clinical situation in hepatocellular carcinoma [259].

## 8. Summary

This review gives a systematic overview on CRPV-based rabbit animal models, starting with the historical development, the association with human papillomaviruses (HPV) and the generation and use of various RPV tumor cell lines. The particular focus is set on the VX2 tumor and the tumor models developed from it in the NZW rabbit.

With the identification of the CRPV in benign and malignant rabbit tumors and its potency for transformation of infected cells, an orthotopic animal model system was established, which has been used for almost 90 years to answer experimental questions in the fields of virology, immunology and oncology. The discovery of high-risk HPV strains as major drivers for transformation of normal into malignant cells and the homology between CRPV and HPV has led enormously to the development of preventive antiviral strategies, has broadened the understanding of the viral life cycle in general, and has contributed to the identification of mechanisms and genes involved in tumorigenesis.

The isolation of CRPV-infected tumor cell lines and the propagation of VX2 cells over several decades by passaging tumor tissue through the muscle of donor rabbits has laid the experimental foundation for the detailed characterization of tumor growth, such as infiltration, neo-angiogenesis and metastasis. During passage through rabbits, the VX2 cells lost their virulence but retained the genetic profile of the CRPV genome in the same manner as observed in HPV+ human HNSCC. This enabled the generation of VX2 cell lines that can be used in vitro and in vivo to experimentally address questions about HPV-associated tumor progression in humans.

The ability of VX2 cells to form metastatic tumors in different anatomical regions and tissues allows the testing and evaluation of surgical procedures, intervention radiological techniques, chemotherapies and targeted delivery strategies, including a realistic assessment of the risk for metastasis. The feasibility of orthotopic transplantation of VX2 tumor cells in immunocompetent rabbits and the potential of solid VX2 tumors for lymphogenic metastasis offer decisive experimental advantages. The fully functional immune system enables the development of immune-based tumor therapies, such as cytokine-based therapies or approaches with checkpoint inhibitors and their evaluation regarding efficacy, side effects and clinical safety. Effects of the tumor microenvironment on tumor growth and therapy-dependent tumor evasion strategies can be assessed in vivo under realistic conditions. 

## 9. Information According to the PRISMA Checklist

The following overview of RPV-based carcinoma models is based on the guidelines of the Preferred Reporting Items for Systematic reviews and Meta-Analyses (PRISMA) Statement 2020 for systematic reviews. [260] The PubMed database https://pubmed.ncbi.nlm.nih.gov/advanced/ was searched for the following terms up to June 2024: (cottontail rabbit papillomavirus, 320 matches); ((VX2), 1420 matches); ((VX2) AND (animal model), 922 matches); ((HNSCC) AND (animal), 769 matches); ((oral papillomavirus) AND (rabbit), 49 matches). Only articles in English were considered. For the evaluation of individual aspects of the RPV models in comparison with the clinical situation, the PubMed search was expanded to include corresponding publications on human HNSCCs. Whenever possible, the work of the first describers of an animal model or method was listed.

## Figures and Tables

**Figure 1 viruses-16-01722-f001:**
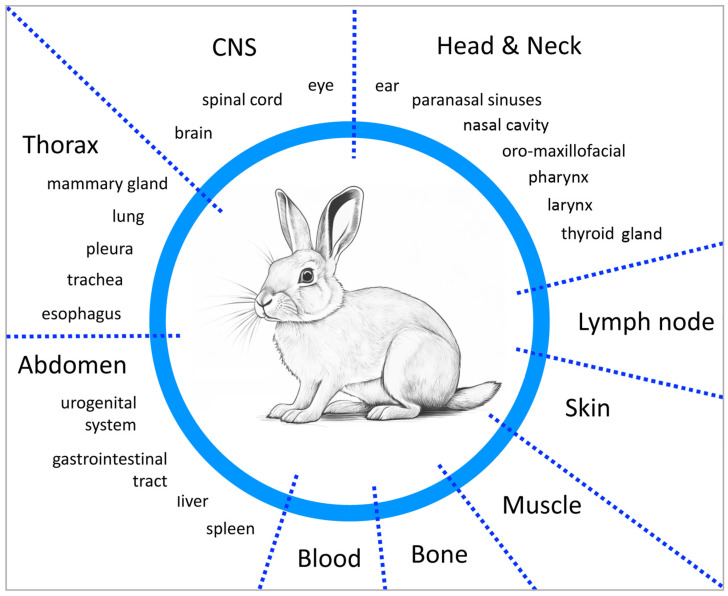
Depiction of VX2 tumor rabbit models grouped according to body regions and organs.

**Figure 2 viruses-16-01722-f002:**
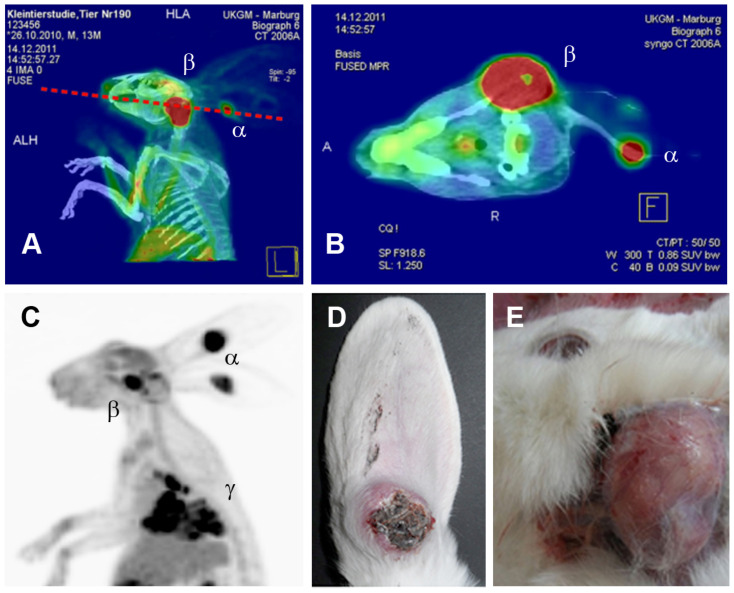
Metastatic spread of VX2 tumors in rabbits. (**A**,**B**) PET-CT analyses showing a rabbit monoauricular VX2 tumor (α) and a lymphatic metastasis in the regional sentinel lymph node (β). (**C**) PET analysis of a rabbit with biauricular VX2 tumors (α) that has developed regional lymph node (β) and distant lung (γ) metastases. Macroscopic views on (**D**) primary auricular VX2 tumor and (**E**) lymph node metastases. Figure adapted from Rossmann et al. 2014 [81].

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
