# Peer review of "Cottontail Rabbit Papillomavirus (CRPV) Related Animal Models for Head and Neck Cancer Research: A Comprehensive Review of the Literature"

_viruses, 2024, doi:10.3390/v16111722_

Round 1
Reviewer 1 Report
Comments and Suggestions for Authors
Review on Bette and Mandic Cottontail rabbit papillomavirus (CRPV) related animal models for head and neck cancer research: a comprehensive review of the literature
1. Please, check PRISMA checklist for reviews and adapt your review to the relevant parts.
2. The title is focused on head and neck cancer research, though 3 pages are about out of the head and neck area models and the main focus is on VX2 type, which fills about the 80% of the review. Title is not fully reflecting the content.
3. The content otherwise is detailed and tries to cover the whole area. The content is logicaly structured.
Author Response
Remark 1: Please, check PRISMA checklist for reviews and adapt your review to the relevant parts.
Response to 1: We have now adapted the manuscript according the guideline of PRISMA 2020 Checklist. For this purpose, we have added a new paragraph at the end of the manuscript, in which the database used, the search method and exclusion criteria are listed. This paragraph reads: “The following overview of RPV-based carcinoma models is based on the guidelines of the Preferred Reporting Items for Systematic reviews and Meta-Analyses (PRISMA) Statement 2020 for systematic reviews. The PubMed database https://pubmed.ncbi.nlm.nih.gov/advanced/ was searched for the following terms up to June 2024: (cottontail rabbit papillomavirus, 320 matches); ((VX2), 1420 matches); ((VX2) AND (animal model), 922 matches); ((HNSCC) AND (animal), 769 matches); ((oral papillomavirus) AND (rabbit), 49 matches). Only articles in English were considered. For the evaluation of individual aspects of the RPV models in comparison with the clinical situation, the PubMed search was expanded to include corresponding publications on human HNSCCs. Whenever possible, the work of the first describers of an animal model or method was listed.”
Remark 2: The title is focused on head and neck cancer research, though 3 pages are about out of the head and neck area models and the main focus is on VX2 type, which fills about the 80% of the Review. Title is not fully reflecting the content.
Response to 2: We have expanded the title accordingly and now call the focus of the VX2 tumor cells. The title now reads: “The cottontail rabbit papillomavirus associated VX2 carcinoma as an animal model for head and neck cancer research: A comprehensive review of the literature.”

Reviewer 2 Report
Comments and Suggestions for Authors
This review presents a comprehensive literature analysis of a unique and fascinating model of rabbit squamous cell carcinoma cell lines derived from CRPV-induced tumors. It focuses on their application in rabbit models for human head and neck tumors. While the manuscript is well-written and highly informative, several clarifications would enhance its value, especially for readers outside the CPPV field:
Major Points:
1. Line 134: Two rabbit papillomaviruses should be referred to as independent species based on the current ICTV classification: Kappapapillomavirus 1 (ROPV) and Kappapapillomavirus 2 (CRPV). The changing nomenclature and variation of manes used in the literature over time can confuse an inexperienced reader. Including a few sentences that summarize the chronology of name variations, and the equivalents would benefit readers.
2. Has ROPV ever been shown to cause any carcinomas aside from warts? If not, this should be clearly stated. The title in Line 155 should be modified from “Induction of RPV-based carcinomas” to “Induction of CRPV-based carcinomas.”
3. Line115. The section should also mention several other known viral genes, such as E10 (formerly E8) and E8^E2 of which many are generated by splicing of primary viral transcripts.
4. Line 200: Despite several early contradictory results, it was concluded that the CRPV genome in VX cell lines is integrated into the host chromosome, resulting in a unique expression pattern in these two cell lines (Georges et al., 1984, J. Virol. 51:530-8; Nasseri and Wettstein, 1984, J. Virol. 51:706-712).
5. The authors may also mention two described VX2 subclones, VX2T and VX2R (Georges et al., 1985, J. Virol. 55:246-250), even though their use in subsequent studies is limited.
6. These cells are maintained for an incredibly long time. A brief description of the laboratory procedures related to their maintenance both in vitro and in vivo, including any published protocols and their current availability, would be of great interest to the readers.
7. The use of VX cell lines in other tumor models unrelated to head and neck-related cancers, seems a little bit excessive (lines 242-405) and could be simplified or summarized in the table. Alternatively, the head and neck sections should be placed first as it is the primary topic of the presented review.
Minor Points: All Latin names should be italicized (e.g., Lines 64, 139, 143…).
Author Response
Major Points:
Remark 1: Line 134: Two rabbit papillomaviruses should be referred to as independent species based on the current ICTV classification: Kappapapillomavirus 1 (ROPV) and Kappapapillomavirus 2 (CRPV). The changing nomenclature and variation of manes used in the literature over time can confuse an inexperienced reader. Including a few sentences that summarize the chronology of name variations, and the equivalents would benefit readers.
Response to 1: We have included this point and inserted the following information in the text: „The nomenclature of these two independent virus species is often used unclearly and inconsistently in the literature, and should be based on the current International Committee on Taxonomy of Viruses (ICTV) classification. A detailed publication on the origin and current nomenclature is provided by Bernard and colleagues (Bernard et al. 2010)”.
New literature added to the reference list:
Bernard HU, Burk RD, Chen Z, van Doorslaer K, zur Hausen H, de Villiers EM. Classification of papillomaviruses (PVs) based on 189 PV types and proposal of taxonomic amendments. Virology. 2010 May 25;401(1):70-9. doi: 10.1016/j.virol.2010.02.002. Epub 2010 Mar 5. PMID: 20206957; PMCID: PMC3400342.
Remark 2: Has ROPV ever been shown to cause any carcinomas aside from warts? If not, this should be clearly stated. The title in Line 155 should be modified from “Induction of RPV-based carcinomas” to “Induction of CRPV-based carcinomas.”
Response to 2: We changed the title and modified the text as follows:
“The host specificity of the two virus species for domestic rabbits makes them ideal candidates for controlled animal experiments. In sharp contrast to CRPV, to the best of our knowledge, ROPV has not been unequivocally linked to malignant transformation. However, experimentally, it was demonstrated that ROPV E6, E7 and E8 can act as oncogenes since cells transfected with these viral proteins lead to tumors in athymic mice (Hu et al., 2004)”
New literature added to the reference list:
Hu J, Cladel NM, Budgeon LR, Christensen ND. Characterization of three rabbit oral papillomavirus oncogenes. Virology. 2004 Jul 20;325(1):48-55. doi: 10.1016/j.virol.2004.04.024. PMID: 15231385.
Remark 3: Line115. The section should also mention several other known viral genes, such as E10 (formerly E8) and E8^E2 of which many are generated by splicing of primary viral transcripts.
Response to 3: We have revised this section accordingly and replaced the former section: “The sequencing analysis revealed that CRPV consists of a double-stranded DNA (dsDNA) 7863 bp long, which codes for a total of 7 genes. These genes can be divided into two main categories consisting of 5 early (E) and 2 late (L) genes. The 2 genes of the (L) category code for the main capsid protein of the outer envelope (L1) and an additional capsid protein (L2) which plays a role in the packaging of the viral DNA and penetration into the host cell. The genes of the (E) category, consisting of E1, E2, E4, E5, E6 and E7, cause viral replication, control the viral life cycle and are promoting the transformation of infected cells, which leads to the formation of papilloma.”
By the following section:
“CRPV consists of a double-stranded DNA (dsDNA) with a size of 7868 bp. CRPV genes can be divided into two main categories consisting of 9 early (E) and 2 late (L) genes. The 2 genes of the (L) category code for the major capsid protein (L1) which forms the majority of the viral the capsid that surrounds and protects the viral genome and an additional minor capsid protein (L2), which plays a role in the packaging of the viral DNA and penetration into the host cell. The genes of the (E) category, consist of E1, E2, E4, E5, LE6, SE6, E7, E8^E2 (formerly E9^E2C) and E10 (formerly E8). E1 and E2 are involved in replication and gene regulation and are required for infection. LE6 and E7 have a major role in cellular transformation. E10 and SE6 promote tumor growth and E4 is involved in DNA synthesis and expression of L1 capsid proteins. E8^E2 represses transcription of the infected host cells whereas E5 exhibits a mutation in its start codon and does not appear to be functional (Cladel et al., 2019).”
New literature added to the reference list:
Cladel NM, Peng X, Christensen N, Hu J. The rabbit papillomavirus model: a valuable tool to study viral-host interactions. Philos Trans R Soc Lond B Biol Sci. 2019 May 27;374(1773):20180294. doi: 10.1098/rstb.2018.0294. PMID: 30955485; PMCID: PMC6501911.
Remark 4: Line 200: Despite several early contradictory results, it was concluded that the CRPV genome in VX cell lines is integrated into the host chromosome, resulting in a unique expression pattern in these two cell lines (Georges et al., 1984, J. Virol. 51:530-8; Nasseri and Wettstein, 1984, J. Virol. 51:706-712).
Response to 4: Thank you for this suggestion. We included this sentence accordingly by replacing the following statement: “In both tumor lines, copies of the CRPV genome are found as free monomers or oligomers [58-62] and the continuous expression of these free or integrated CRPV copies is essentially responsible for the malignant transformation of CRPV-infected cells.”
It now reads.
“Despite several early contradictory results, it was concluded that the CRPV genome in VX cell lines is integrated into the host chromosome, resulting in a unique expression pattern in in the VX2 and VX7 tumor cells.
Remark 5: The authors may also mention two described VX2 subclones, VX2T and VX2R (Georges et al., 1985, J. Virol. 55:246-250), even though their use in subsequent studies is limited.
Response to 5: We have inserted the following sentence: “Furthermore, contradGeorges and coworkers used the VX2 tumor to generate two cell lines, VX2T and VX2R, and noticed loss of transplantability especially for the VX2R cell line (Georges et al., 1985)”
New literature added to the reference list:
Georges E, Breitburd F, Jibard N, Orth G. Two Shope papillomavirus-associated VX2 carcinoma cell lines with different levels of keratinocyte differentiation and transplantability. J Virol. 1985 Jul;55(1):246-50. doi: 10.1128/JVI.55.1.246-250.1985. PMID: 2409299; PMCID: PMC254922.
Remark 6: These cells are maintained for an incredibly long time. A brief description of the laboratory procedures related to their maintenance both in vitro and in vivo, including any published protocols and their current availability, would be of great interest to the readers.
Response to 6: We added the following information: “Propagation of VX2 cells was achieved by passaging tumor tissue, which contains VX2 tumor cells together with their microenvironment (tumor stroma). This procedure is described in detail by van Es and colleagues (van Es, 2000) and is considered crucial for the VX2 tumor to survive for several decades.” Literature was already in the reference list.
Remark 7: The use of VX cell lines in other tumor models unrelated to head and neck-related cancers, seems a little bit excessive (lines 242-405) and could be simplified or summarized in the table. Alternatively, the head and neck sections should be placed first as it is the primary topic of the presented review.
Response to 7: We changed the heading to “Rabbit VX2 tumor models in the head and neck area” directly followed by the “Head and Neck” section. We then included another heading that reads “Other rabbit VX2 tumor models” followed by the respective sections for CNS, Thorax, Abdomen, Bone, Skin, Muscle and Blood
Minor Points: All Latin names should be italicized (e.g., Lines 64, 139, 143…).
Response to minor points: corrected. We have also changed the name of genes to italic.

Reviewer 3 Report
Comments and Suggestions for Authors
Bett and Mandic briefly review the history of CRPV, the genomic organization and rabbit cancers related to this virus. Most of the manuscript is focused on a comprehensive description of the many different cancer models that have been reported based on implantation/inoculation of VX2 carcinoma cells at various different anatomical sites. While there is slightly more focus on the head and neck this aspect of the review was somewhat less prevalent than I was expecting.
The review was well written and well referenced. I would support publication with some minor corrections the text and some additional details in certain areas.
Minor points:
1) Figures 2 and 3 seem somewhat low in terms of resolution
2) add more content on the frequency of HPV+ head and neck cancer and the current "epidemic" given the focus of the review. In the USA, the number of annual cases of male HPV+ HNSCC now is higher than cervical cancer. It is also estimated that this will continue until ~2045 despite the vaccine. This is a problem that is not going away anytime soon.
3) I would like to see a little more content and similarities in oncogenic transformation by CRPV vs HPV. Just a few sentences would be fine. Do the CRPV oncogenes target Rb and TP53 like HPV E7 and E6? Maybe site another review on this as well as drawing similarities between mechanisms?
4) Maybe I missed it, but it would be nice to have a short section on similarities in CRPV induced cancer pathology compared to human HPV+ HNSCC/OPSCC. Things like T cell/B cell/lymphocyte infiltration, p16 positivity, lymph node involvement etc, which are commonly scene in human HPV+ OPSCC.
5) I believe NewZealand rabbits weigh at least several kgs vs mice at 20g. As a potential problem/caveat that should be mentioned, the authors should point out the size difference in terms of such numbers. Importantly, animal weight/size also dictates large increases in drug treatment amounts, which could be prohibitively expensive for costly infusible biologics like antibodies. The authors should also comment on the availability of immune checkpoint inhibitor therapies that work in rabbits. Not sure the human or mouse versions of anti-CTLA4 or PD-1/PD-L1, etc will work on rabbit immune cells. I know they have similar problems in woodchuck models of HBV infection/liver cancer and immune modification treatments (PMID:34163570) and woodchucks are probably smaller than rabbits.
Comments on the Quality of English LanguageSuggested text corrections:
1) Sarcoma not sarkoma (line71)
2) divide the summary paragraph into at least 2 or more likely 3 paragraphs. The topics here change often enough that their needs to be some paragraph breaks to enhance readability.
3) PVs are non-enveloped. It's a little confusing to call the L main capsid proteins (plural) envelope on line113. Calling it the outer envelope also suggests that there is an inner envelope.
Author Response
Response to Reviewer 3:
Remark 1: Figures 2 and 3 seem somewhat low in terms of resolution
Response to 1: The resolution has been adjusted. The high-resolution images were submitted to the publisher.
Remark 2: add more content on the frequency of HPV+ head and neck cancer and the current "epidemic" given the focus of the review. In the USA, the number of annual cases of male HPV+ HNSCC now is higher than cervical cancer. It is also estimated that this will continue until ~2045 despite the vaccine. This is a problem that is not going away anytime soon.
Response to 2: We added the following content: “Barsouk et al. pointed out a growing prevalence of HPV associated HNSCC, predicting that HPV+ HNSCC will surpass tobacco as a risk factor for HNSCC (Barsouk et al., 2023). Therefore, papillomavirus driven tumor models, such as the CRPV associated VX2 carcinoma, could become more and more significant to experimentally address new therapeutic approaches prior to use in HNSCC patients.”
New literature added to the reference list:
Barsouk A, Aluru JS, Rawla P, Saginala K, Barsouk A. Epidemiology, Risk Factors, and Prevention of Head and Neck Squamous Cell Carcinoma. Med Sci (Basel). 2023 Jun 13;11(2):42. doi: 10.3390/medsci11020042. PMID: 37367741; PMCID: PMC10304137.
Remark 3: I would like to see a little more content and similarities in oncogenic transformation by CRPV vs HPV. Just a few sentences would be fine. Do the CRPV oncogenes target Rb and TP53 like HPV E7 and E6? Maybe site another review on this as well as drawing similarities between mechanisms?
Response to 3: We added the following content: “In particular, studies on the viral oncogenes E6 and E7, which also play a fundamental role in HPV diseases, characterize their impact in cellular transformation. Interestingly, while the CRPV oncoproteins E6 and E7, as observed for HPV, are required for oncogenic transformation of the cell, there are some differences between these CRPV and HPV oncoproteins. As known for HPV E7, CRPV E7 also inhibits the retinoblastoma (RB) (Haskell et al., 1993). However, in sharp contrast to HPV, CRPV E6 does not bind p53 and also cannot induce its degradation (Harry and Wettstein, 1996). Instead, Du et al. demonstrate binding of CRPV E6 (LE6=long E6 and SE6=short E6) to another tumor suppressor, hDlg/SAP97, which could explain its oncogenic potential analogous to p53 inhibition (Du et al., 2005)”.
New literature added to the reference list:
Haskell KM, Vuocolo GA, Defeo-Jones D, Jones RE, Ivey-Hoyle M. Comparison of the binding of the human papillomavirus type 16 and cottontail rabbit papillomavirus E7 proteins to the retinoblastoma gene product. J Gen Virol. 1993 Jan;74 (Pt 1):115-9. doi: 10.1099/0022-1317-74-1-115. PMID: 8380832.
Harry JB, Wettstein FO. Transforming properties of the cottontail rabbit papillomavirus oncoproteins Le6 and SE6 and of the E8 protein. J Virol. 1996 Jun;70(6):3355-62. doi: 10.1128/JVI.70.6.3355-3362.1996. PMID: 8648665; PMCID: PMC190206.
Du M, Fan X, Hanada T, Gao H, Lutchman M, Brandsma JL, Chishti AH, Chen JJ. Association of cottontail rabbit papillomavirus E6 oncoproteins with the hDlg/SAP97 tumor suppressor. J Cell Biochem. 2005 Apr 1;94(5):1038-45. doi: 10.1002/jcb.20383. PMID: 15669058.
Remark 4: Maybe I missed it, but it would be nice to have a short section on similarities in CRPV induced cancer pathology compared to human HPV+ HNSCC/OPSCC. Things like T cell/B cell/lymphocyte infiltration, p16 positivity, lymph node involvement etc, which are commonly scene in human HPV+ OPSCC.
Response to 4: Thank you for this point of criticism. We have revised the relevant section here and replaced the section: “Furthermore, these rabbit tumors show a pronounced tendency for lymphogenic metastasis [97,186], which is comparable to metastasis formation in HNSCC [187]. In addition, VX2 tumors show a low degree of keratinization similar to HNSCCs (Fig 3).”
by the following section:
“There are several similarities between the CRPV driven VX2 carcinoma and human HPV+ HNSCC. Clinically, both tumors primarily exhibit a lymphatic metastatic spread to locoregional lymph nodes (REFs 97,186), which is one major aspect why the auricular VX2 carcinoma was initially considered a suitable model system for human HNSCC. Histopathologically, both tumors, the CRPV associated VX2 carcinoma and HPV+ human HNSCCs, exhibit a low differentiation level (basaloid) that typically is seen in papillomavirus driven tumors. This is also highlighted by the reduced keratinization level seen in Fig.3. Furthermore, in our previous studies, we demonstrated tumor-infiltrating leukocytes (TILs) to correlate with therapy response of VX2 carcinomas (Rossmann et al, 2014), which is in accordance with observations in human HPV+ HNSCC (Yin et al., 2023).“
New literature added to the reference list:
Yin LX, Rivera M, Garcia JJ, Bartemes KR, Lewis DB, Lohse CM, Routman DM, Ma DJ, Moore EJ, Van Abel KM. Impact of Tumor-Infiltrating Lymphocytes on Disease Progression in Human Papillomavirus-Related Oropharyngeal Carcinoma. Otolaryngol Head Neck Surg. 2023 Sep;169(3):539-547. doi: 10.1002/ohn.249. Epub 2023 Jan 30. PMID: 36939471.
Remark 5: I believe New Zealand rabbits weigh at least several kgs vs mice at 20g. As a potential problem/caveat that should be mentioned, the authors should point out the size difference in terms of such numbers. Importantly, animal weight/size also dictates large increases in drug treatment amounts, which could be prohibitively expensive for costly infusible biologics like antibodies. The authors should also comment on the availability of immune checkpoint inhibitor therapies that work in rabbits. Not sure the human or mouse versions of anti-CTLA4 or PD-1/PD- L1, etc. will work on rabbit immune cells. I know they have similar problems in woodchuck models of HBV infection/liver cancer and immune modification treatments (3) PMID:34163570) and woodchucks are probably smaller than rabbits PVs are non-enveloped. It's a little confusing to call the L main capsid proteins (plural).
Response to 5: The information on the capsid has been corrected (see also response 3 to Reviewer 2)
To date, there are 2 studies on checkpoint inhibitors and the use of antibodies to modulate the checkpoint signaling pathways. In the study by Roy et al on ocular herpes simplex virus (HPV) infection in rabbits, it is shown that monoclonal antibodies can block the PD-1 and LAG-3 signaling pathways. In the study by Berz et al, histological staining of anti-PD-1, anti-PD-L1 and anti-CTLA-4 antibodies in VX2 tumors of the liver are shown. The commercial source of the antibodies is mentioned in the study. We have now included this information under the section “Ear” in the paragraph about innovative cancer treatment strategies.
It reads: “One way of triggering or enhancing an anti-tumor response is the targeted administration of immunomodulatory drugs such as pro-inflammatory cytokines or checkpoint inhibitors such as programmed cell death protein-1 (PD-1), programmed cell death protein-1 ligand (PD-L1) and cytotoxic T-lymphocyte antigen 4 (CTLA-4). Such immunomodulators can be either which are either administered intravenously or packaged in microsphere carrier systems and introduced into the tumor area via the tumor-bearing vessel. Today, there are commercially available monoclonal antibodies that functionally block rabbit PDL-1 and LAG-3 signaling pathways (Roy et al, 2019) or can be used for immunohistological detection of PD-1, PDL-1 and CTLA-4 in rabbits (Berz, 2022)”.
New literature added to the reference list:
Roy S et al. Blockade of PD-1 and LAG-3 Immune Checkpoints Combined with Vaccination Restores the Function of Antiviral Tissue-Resident CD8+ TRM Cells and Reduces Ocular Herpes Simplex Infection and Disease in HLA Transgenic Rabbits. J Virol. 2019 Aug 28;93(18):e00827-19. doi: 10.1128/JVI.00827-19. PMID: 31217250; PMCID: PMC6714801
Berz AM. Impact of Chemoembolic Regimen on Immune Cell Recruitment and Immune Checkpoint Marker Expression following Transcatheter Arterial Chemoembolization in a VX2 Rabbit Liver Tumor Model. J Vasc Interv Radiol. 2022 Jul;33(7):764-774.e4. doi: 10.1016/j.jvir.2022.03.026. Epub 2022 Mar 26. PMID: s; PMCID: PMC9344951.
Suggested text corrections:
Remark 1: Sarcoma not sarkoma (line71)
Response to 1: Corrected
Remark 2: divide the summary paragraph into at least 2 or more likely 3 paragraphs. The topics here change often enough that their needs to be some paragraph breaks to enhance readability.
Response to 2: We agree with the reviewer that the summary paragraph is too detailed and in parts redundant to the text. We have therefore revised and shortened it. Individual aspects have been included in the text under paragraph “Rabbit as a model organism“.
Remark 3: envelope on line113. Calling it the outer envelope also suggests that there is an inner envelope.
Response to 3: The information on the L genes has been corrected (see also response 3 to Reviewer 2)
